# DBU-Net: Dual branch U-Net for tumor segmentation in breast ultrasound images

**Payel Pramanik**[1], **Rishav Pramanik**[1], **Friedhelm Schwenker**[2]*, **Ram Sarkar**[1]

**1** Department of Computer Science and Engineering, Jadavpur University, Kolkata, India, **2** Institute of Neural Information Processing, Ulm University, Ulm, Germany

* friedhelm.schwenker@uni-ulm.de

**Data Availability Statement:** Publicly available dataset is analyzed in this study. The data can be found here: [URL: https://www.kaggle.com/datasets/aryashah2k/breast-ultrasound-images-dataset (accessed on 20th February, 2023)].

## Abstract

Breast ultrasound medical images often have low imaging quality along with unclear target boundaries. These issues make it challenging for physicians to accurately identify and outline tumors when diagnosing patients. Since precise segmentation is crucial for diagnosis, there is a strong need for an automated method to enhance the segmentation accuracy, which can serve as a technical aid in diagnosis. Recently, the U-Net and its variants have shown great success in medical image segmentation. In this study, drawing inspiration from the U-Net concept, we propose a new variant of the U-Net architecture, called DBU-Net, for tumor segmentation in breast ultrasound images. To enhance the feature extraction capabilities of the encoder, we introduce a novel approach involving the utilization of two distinct encoding paths. In the first path, the original image is employed, while in the second path, we use an image created using the Roberts edge filter, in which edges are highlighted. This dual branch encoding strategy helps to extract the semantic rich information through a mutually informative learning process. At each level of the encoder, both branches independently undergo two convolutional layers followed by a pooling layer. To facilitate cross learning between the branches, a weighted addition scheme is implemented. These weights are dynamically learned by considering the gradient with respect to the loss function. We evaluate the performance of our proposed DBU-Net model on two datasets, namely BUSI and UDIAT, and our experimental results demonstrate superior performance compared to state-of-the-art models.

## Introduction

Breast cancer poses a significant threat to women's health, accounting for 11.7% of the total cancer incidence rate globally and surpassing lung cancer to become the most common cancer [1]. Early detection plays a crucial role in providing timely clinical decisions, treatments, and rehabilitation plans, ultimately reducing mortality. Breast cancer is commonly detected through methods such as physical examination, imaging techniques like mammography, ultrasound, and breast magnetic resonance imaging (MRI), as well as biopsy [2]. However, physical examination can be challenging to differentiate between malignant and benign lesions and may require experience. Biopsy is considered the gold standard for determining the nature of a lesion but can be a painful and inconvenient process, sometimes requiring multiple attempts.

**Funding:** The author(s) received no specific funding for this work.

**Competing interests:** The authors have declared that no competing interests exist.

Access to biopsy labs may also be limited in remote or low-resource areas, leading to delays in diagnosis.

To reduce the need for unnecessary biopsies and uncomfortable physical exams, ultrasound imaging is an attractive alternative to mammography and MRI due to its improved sensitivity, lack of radiation, low cost, and widespread availability. However, ultrasound imaging has several limitations, including low contrast, poor resolution, fuzzy edges due to noise such as speckle, acoustic shadowing, and indistinct surrounding tissue. Thus, breast tumor diagnosis in ultrasound images remains time-consuming, challenging, and subjective for radiologists. To simplify this process, computer-aided diagnostic (CAD) systems have been developed, providing reliable results and streamlining operations [3]. These methods are known to be cost effective and time saving. It should be very well noted that the occurrence of breast cancer is much more of a concern in low or middle income countries. In most of these countries, the healthcare infrastructure is often not very well developed in comparison with developed countries. This poses a significant challenge to the majority of the population being vulnerable to contract these potentially fatal diseases like breast cancer [4]. CAD systems can serve as valuable tools for the medical industry, enabling cost-effective solutions across various domains of healthcare [5–10].

Segmentation of cancer region in breast images is one of paramount steps to identify the presence of any lesion. Medical image segmentation aims to identify and isolate specific areas within images that hold significant medical importance. By doing so, relevant areas or regions of interest can be highlighted, which can be used as a reliable basis for clinical diagnosis and pathology research [11, 12]. However, medical image analysis poses several challenges such as variations in texture, shape, and individual differences, which make manual annotation a prevalent practice in clinical settings. This process is time-consuming and requires specialized expertise. Consequently, there is a growing need for automated segmentation methods that are accurate and reliable. Such methods can reduce the workload of clinical experts and help them to improve their efficiency [11]. In literature, numerous segmentation procedures have been adopted by various researchers. In a typical image segmentation process, an input image is considered and the corresponding segmentation map is expected as the output.

Since the rise of deep learning in the last decade, Convolutional neural network (CNNs) based models have made a remarkable progress in various image segmentation tasks [13–18]. In 2015, Jonathan et al. [19] started exploring the application of CNNs for performing automatic segmentation tasks in an end-to-end manner. They introduced a novel architecture called the fully convolutional neural network (FCN), which gained widespread adoption for image segmentation using CNNs in an end-to-end fashion. However, unlike ImageNet, medical image datasets usually contain highly similar images, making it challenging to extract sufficient context information and receptive fields using FCNs. This can result in poor segmentation performance [20]. To address this, researchers have proposed advanced frameworks to enhance efficiency. One popular approach is U-Net [21], which is based on FCNs, but can extract richer contextual information with more sufficient receptive fields, leading to improved performance in medical image segmentation. U-Net is a widely used network in medical image segmentation due to its ability to extract contextual information through skip connections. The network has an encoder-decoder setup, wherein the encoder downsamples the image to extract features while the decoder utilizes these features (from the encoder) to upsample the output segmentation mask through the help of skip connections. This allows the network to obtain features of different granularity, leading to generate improved segmentation masks. The presence of skip connections facilitates the transfer of low-level to high-level features from the encoder to the decoder, which ultimately leads to an enhanced comprehension of the contextual information. Overall, U-Net's effectiveness in concatenating contextual

information through skip connections has made it a popular choice for medical image segmentation [22, 23].

After the emergence of U-Net, several novel methods have been proposed to improve the performance of the medical image segmentation task. For example, Deep Residual U-Net [24] integrates residual blocks into the encoder and decoder layers, which deepens the network and enhances its performance. Other models, such as RCNN and R2CNN by Alom et al. [25], use recurrent mechanisms to accumulate features. BCDUNet [26] uses bi-directional ConvLSTM instead of skip connections and applies a block of dense convolutions to the bottom encoding layer. Attention mechanisms have also been introduced in skip connections of U-Net [27]. To address the issue of blending semantically distinct features in the regular skip connections of U-Net, U-Net++ [28] enhances the standard skip connections with nested and dense skip connections. U-Net++ implements a deep monitoring mechanism that allows for the removal of dense network structures, thereby increasing the model's flexibility. Apart from making changes to the architecture of neural networks, researchers have also considered and investigated the possibility of modifying the size of kernels used in these networks [29].

While these models have achieved state-of-the-art performance in some tasks of medical image segmentation, they focus mainly on modifying contextual feature extraction concepts, and passing information between the encoder and the decoder. This approach may cause misclassification on the pixel of boundary areas, since they do not consider the simultaneous extraction of spatial and contextual information. Each of these networks, for instance UNet++, R2CNN among many others exploit certain characteristics of features extracted by the encoder that are further used to generate the segmentation mask. Among this pool of deep segmentation networks, there are very less methods that focus on enriching the encoded features through the use of edge detection. From a visual perspective, the edges for a particular object can be used for localizing the object. Inspired by this idea, in this work, we present an end-to-end deep neural network based segmentation network, called DBU-Net, which fuses information from edges and original images. The encoded information learned by each of the branches fuses Roberts edge information with the encoding from original images. The ultimate goal is to localize the context in order to produce a better segmentation mask, which can be served as a support tool to the medical professionals for a cheap, robust and fast diagnosis of breast cancer.

The main contributions of this paper are summarized as follows:

- Our paper introduces DBU-Net, a novel approach for precise medical image segmentation. Unlike the vanilla U-Net, we place greater emphasis on the feature extraction process in the encoder path and present a dual-encoder model.

- Our proposed approach involves utilizing two separate input paths for the encoding process. One of these branches incorporates the original image, and the other path uses the Roberts edge information obtained from the original image.

- The dual-branch encoding strategy is employed to enrich the semantic information within the latent space, utilizing cross-learning methods. To facilitate cross-learning, a weighted addition mechanism is utilized, while the weight is determined based on the loss gradient during the training of the model.

- The performance of the proposed method is evaluated on two breast cancer datasets, namely BUSI and UDIAT. The outcomes are highly encouraging, with IoU scores of 74.34% and 77.46%, and Dice scores of 85.28% and 87.28%, obtained on BUSI and UDIAT datasets, respectively.

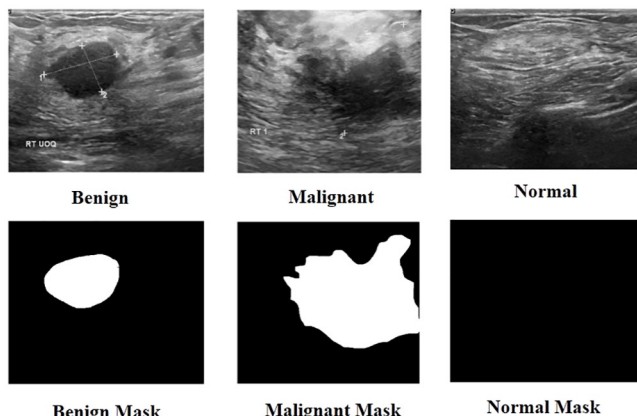

**Fig 1. Examples of breast ultrasound images and associated masks in the BUSI dataset.**

The remainder of the paper is structured as follows. First, the proposed method and the dataset used are described. Next, the experimental results and analysis of the proposed method and the discussion of the results are presented. Finally, we conclude our work, and state some limitations and future extension possibilities.

## Materials and methods

In this section, we first provide a comprehensive explanation of the dataset employed for the experimentation, followed by a thorough discussion about our proposed model.

### Dataset description

In this study, the proposed technique is trained and evaluated using the BUSI [30] dataset. In 2018, the BUSI dataset was gathered for 600 female patients aged between 25 and 75 years. The baseline data included breast ultrasound images, with an average image size of 500 × 500 pixels, in PNG format. The dataset contains 780 images along with ground truth masks that are categorized into three classes: normal, benign, and malignant. However, we have considered the benign and malignant images for the current task. Fig 1 illustrates sample images along with the masks taken from the dataset.

Table 1 shows the distribution of BUSI images across the three classes.

### Data pre-processing and partitioning

In this section, we discuss the data pre-processing and partitioning techniques that we have applied to the images of the BUSI dataset. A total of 647 images from both the benign and malignant classes are taken into consideration. Since normal images do not have label masks,

**Table 1. The number of images in each of the three kinds of breast cases.**

| Class | No. of images |
|---|---|
| Normal | 133 |
| Benign | 437 |
| Malignant | 210 |
| **Total** | **780** |

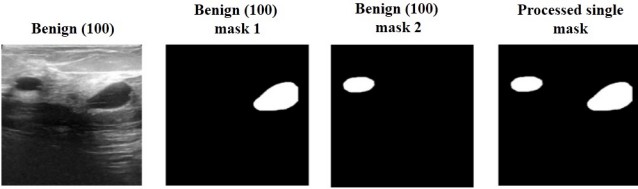

**Fig 2. A sample image with multiple masks, and the single mask obtained after merging them.**

133 instances of normal images are excluded from the current task. To address the issue of non-uniform sizes in the original BUSI images, we resize all images to a uniform size of $256 \times 256$ pixels. To optimize computational resources, we perform image normalization by scaling the pixel values from the original range of 0 to 255 to a new range of 0 to 1. This is achieved by dividing each pixel value by the maximum possible pixel value, which is 255. Further, there are 17 cases (16 benign cases and 1 malignant case) with multiple masks, i.e., here for each image there are multiple masks but all belong to the same class. In these cases, we merge the associated masks together to obtain a single mask. One image of such type is shown in Fig 2.

Additionally, we divide the dataset into training and test sets using a five-fold cross-validation approach. The 5-fold cross-validation method involves dividing the dataset into 5 equal-sized groups, referred to as folds. During experimentation, the learning model is trained on 4 out of the 5 folds, and the remaining fold is used for testing. This process is repeated for all possible combinations of training and testing folds.

## Overview of the proposed method

U-Net architecture consists of two components: 1) an encoder (contracting path), and 2) a decoder (expansive path). The basic structure considers mapping from the encoder to the decoder in a discretely continuous fashion. While, it must be noted that we typically use CNNs only in a U-Net. In this work, inspired by this idea of U-Net, we propose to use two encoding paths consisting of separate inputs. In one of the branches, we use the original image, and the Roberts edge image in the second path. With this dual branch encoding scheme, we mainly try to enrich the semantic information in the latent space in a cross learning fashion. In each level of the encoder, individually the branches consist of two levels of convolutions followed by pooling. Finally, for cross learning, we utilize a weighted addition scheme, where the weights are learned considering the gradient w.r.t. the loss. Fig 3 provides a comprehensive view of the entire pipeline.

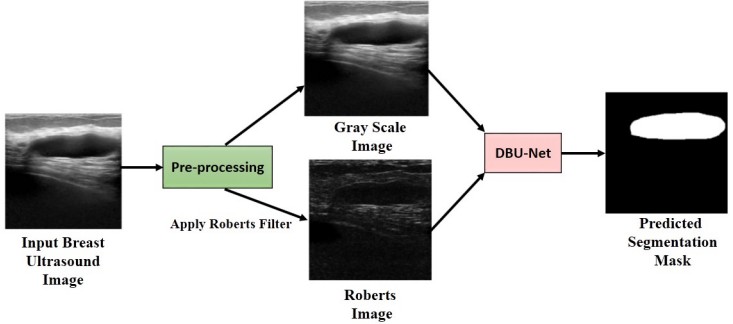

**Fig 3. Overall pipeline of the proposed breast ultrasound image segmentation model.**

## Roberts edge operator

Roberts edge operator is one of the very well known techniques for edge detection in an image. Roberts edge operator consists of two cross operators of size 2 × 2 that are shown in Eqs (1) and (2).

$$\delta_x = \begin{bmatrix} +1 & 0 \\ 0 & -1 \end{bmatrix} \tag{1}$$

$$\delta_y = \begin{bmatrix} 0 & +1 \\ -1 & 0 \end{bmatrix} \tag{2}$$

These operators are convolved with a gray-scale image (say, $I_g$) and generate two gradient images as $G_x$ and $G_y$ as shown in Eqs (3) and (4), respectively.

$$G_x = I_g * \delta_x \tag{3}$$

$$G_y = I_g * \delta_y \tag{4}$$

In Eqs (3) and (4), $*$ is the convolutional operator. The magnitude image corresponding to the gradient images (say, $G$) is calculated using Eq (5).

$$G = \sqrt{G_x^2 + G_y^2} \tag{5}$$

We obtain the final edges using a threshold value (say, $th$). We use the mean of the gradient magnitude values appearing in $G$ to calculate this threshold value. The calculation is shown in Eq (6).

$$th = \frac{1}{MN} \sum_{x=1}^{M} \sum_{y=1}^{N} G(x, y) \tag{6}$$

In Eq (6), $M$ and $N$ stand for the height and width of the input image, respectively. Finally, an edge image (say, $I_e$) is obtained using Eq (7). Pixels with magnitude values above the threshold value ($th$) are considered as edge pixels (white in the output), while those below the threshold are considered as background pixels (black in the output).

$$I_e(x, y) = \begin{cases} edge\ pixel & : G(x, y) > th \\ background\ pixel & : otherwise \end{cases} \tag{7}$$

It applies $\updownarrow_2$ based normalization to merge two filters obtained from the said operators. It should be noted that these images are first normalized in the range [0, 1] and then upscaled to the original resolution. Predominantly, we observe Roberts edge operator to be used in various kinds of computer vision tasks [31, 32]. The major motivation behind utilizing edge images is to improve performance, as they enhance the ability to differentiate between an object and its background. However, on the other hand, this method is very susceptible to noise and thus should be avoided when encountering a noisy image. In this work, we do not deal with very noisy images, therefore, we apply Roberts edge operator.

Here, by using Roberts edge operator, we aim to outline the lesion area for the encoder to approximately highlight the region of interest to extract more informative features. This idea is intuitive and supported by the fact that to capture an ultrasound image, the sound waves are

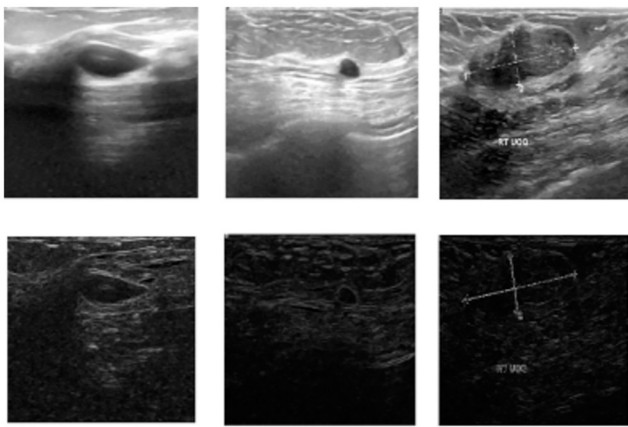

**Fig 4. Sample original images from the BUSI dataset and their corresponding edge-enhanced images after undergoing the Roberts edge operator.** The top row of the dataset shows the original images, while the bottom row displays the corresponding images produced by applying the Roberts edge operator.

transmitted and the echoes are captured. It is quite natural to perceive that an area with abnormality in terms of mass/unnatural growth of cells will have a considerable amount of change in the texture of that region when compared to the surrounding, which has a normal growth. This very fact motivates us to utilize edge images as it can capture the gradient component and can amplify if the gradient is considerably large. This is very much probable in our case. Fig 4 displays both original images taken from the BUSI dataset and the corresponding images after undergoing the Roberts edge operator.

## Dual encoding branch

Fig 1 shows that the images present a significant challenge in terms of segmenting out the lesions from the image. The main objective of this work is to accurately identify the region containing the lesion, but it is visually apparent that the lesion can sometimes appear hollow. This creates a considerable difficulty in distinguishing the boundary regions, which is further compounded by the similarity in texture throughout the image. Therefore, it is crucial to accurately highlight the boundary areas to produce an effective segmentation map. We address this challenge by utilizing Roberts edge information to encode and highlight the regions, thereby improving the quality of the segmentation map. To be precise, a dual branch encoding module considers two inputs for two branches. Each branch has two levels of convolution with $3 \times 3$ filter size. Each of these convolutions is followed by rectified linear unit (ReLU) activation. Following the convolution, for dimension reduction, we pool these features using Maxpooling operation with $2 \times 2$ window size and the stride of the window is equal to (2, 2). This process is followed with dropout of probability 0.2.

We fuse these features in a crosswise manner in order to felicitate cross learning. In simpler terms, we use a crosswise fusion approach to enable the exchange of detailed information between different types of features in the input. This is done by assigning equal weights to the processed feature maps of each type, and then optimizing these weights throughout the network using an appropriate optimizer. This is achieved in accordance with Eq (8), where $F_x$ is the processed feature map of one type, $F_y$ is the other type of input and $W_i$ is the weightage value where $i \in \{x, y\}$. The operators (+) and (·) are simple addition and multiplication respectively. We initially set these values equal to 1. We further optimize these weights by optimizing

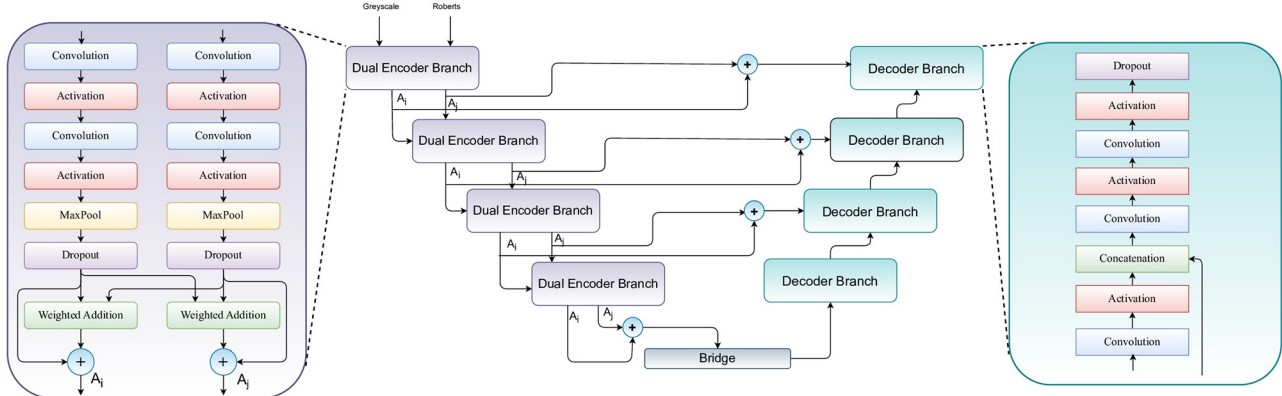

**Fig 5. The architecture of the proposed DBU-Net model used for tumor segmentation in breast ultrasound images.**

this with the entire network with an appropriate optimizer. By adding the original inputs, we preserve the unique qualities of each generated feature map and prevent the two branches from becoming identical. This ensures that both types of features are utilized in every layer of the network. The architecture of the proposed DBU-Net model is shown in Fig 5.

$$F_x = F_x + (W_x \cdot F_x + W_y \cdot F_y) \tag{8}$$

## Decoding: The expansive path

In the expansive path of the network, each step involves increasing the size of the feature map, followed by a $2 \times 2$ convolution that reduces the number of feature channels by half. The resulting feature map is then concatenated with the corresponding cropped feature map from the contracting path. This is necessary because border pixels are lost during convolution. Two $3 \times 3$ convolutions, each followed by a ReLU, are then applied. The final layer of the network uses a $1 \times 1$ convolution to map each feature vector to the desired number of classes.

## Loss function

In the present work, we have used a hybrid loss function which is the combination of focal loss and dice loss.

**Focal loss.** When dealing with class imbalance in datasets, the use of Cross-Entropy loss can introduce bias, as the majority class has a stronger influence on the loss value. Consequently, the model may become overly confident in predicting the majority class samples while neglecting the minority class. To address this issue, Focal loss [33] proves to be more effective in training models on imbalanced datasets. Focal loss tackles the problem by down-weighting or reducing the impact of easy-to-train samples in the loss calculation, thereby prioritizing the harder-to-train samples. This is achieved by introducing a modulating factor, $(1 - P_t)^\gamma$, with tunable focusing parameter $\gamma$, to the Cross-Entropy loss, which directs the learning process towards challenging misclassified samples. The modulating factor is dynamically adjusted and decreases to zero as the model's confidence in the true positive class increases. As a result, $\gamma$ automatically reduces the contribution of easy samples during training, enabling the model to focus more rapidly on the difficult samples. The value of $\gamma$ typically ranges from 0 to 5. The formula of $\alpha$-balanced variant of the focal loss is given in Eq (9), where $P_t$ is the model's prediction probability, $\alpha_t$ is the weighting factor ranges between 0 and 1. In the present work,

we have used the value of $\gamma$ and $\alpha_t$ as 2 and 0.25 respectively.

$$Focal\ loss = -\alpha_t \times (1 - P_t)^{\gamma} \times \log P_t \tag{9}$$

**Dice loss.** The Dice loss is a loss function that directly optimizes the Dice similarity coefficient (DSC), a widely used evaluation metric in segmentation tasks, as described in the subsequent section. Dice loss is calculated according to Eq (10).

$$Dice\ Loss = 1 - DSC \tag{10}$$

**Hybrid loss.** In our approach, we have combined the two distinct loss functions mentioned above, each with its own unique training dynamics, to demonstrate diverse properties. By doing so, we are able to merge the benefits of these loss functions while reducing their drawbacks. The hybrid loss is calculated as per Eq (11).

$$Hybrid\ Loss = Focal\ Loss + Dice\ Loss \tag{11}$$

## Statement of ethical approval

All procedures performed in studies involving human participants were in accordance with the ethical standards of the institutional and/or national research committee and with the 1964 Helsinki Declaration and its later amendments.

## Results and discussion

In this section, initially, we provide an overview of the experimental setup, including information on hyperparameters and the metrics used for evaluating performance. We then delve into our findings and analyze the results we obtained. Finally, we compare our proposed method to other commonly used approaches for breast ultrasound image segmentation.

### Experimental setup

The proposed method is implemented using Python 3.9 and employs various Machine/Deep Learning libraries, including scikit-learn 1.2.2, numpy 1.22.4, pandas 1.4.4, keras 2.11.0, and tensorflow 2.11.0. The experiments are performed on a system equipped with an NVIDIA Tesla T4 GPU, with 15 GB of GPU RAM and 12.7 GB of system RAM.

### Evaluation metrics

This section presents the evaluation metrics utilized to assess the performance of the proposed method described in this paper. The first metric is the DSC index [34], which measures the level of agreement between the original target and the segmented target. We calculate the similarity between two images by measuring the ratio of twice the overlapped area of the images to the total number of pixels in both images. A higher DSC value indicates a better fit between the two images signifying a more accurate segmentation model for breast cancer. The DSC value is calculated according to the Eq (12).

$$DSC = \frac{2 \cdot ||A \cap B||}{||A|| + ||B||} = \frac{2TP}{2TP + FP + FN} \tag{12}$$

In this context, True Positive (TP) refers to the region, where the tumor is present and accurately segmented. False Positive (FP) denotes the area, where the tumor is present but not correctly segmented, while False Negative (FN) indicates the region, where the tumor is absent and remains unsegmented.

Another important metric is the IoU score [34], which is also known as the Jaccard index, is commonly used in image segmentation to measure how much the predicted mask overlaps with the ground truth mask. It calculates the ratio of the number of pixels that are shared by both masks to the total number of pixels present in both masks. This metric is useful because it provides a measure of the quality of the predicted mask by quantifying the level of agreement between it and the ground truth mask. It is calculated in accordance with Eq (13).

$$IoU = \frac{Area\ of\ Intersection}{Area\ of\ Union} = \frac{||A \cap B||}{||A \cup B||} = \frac{TP}{TP + FP + FN} \tag{13}$$

Also, we have considered the overall accuracy for performance evaluation of the proposed method as the percent of pixels from all classes that are accurately classified in the image. It is calculated according to Eq (14).

$$Pixel\ Accuracy = \frac{TP + TN}{TP + TN + FP + FN} \tag{14}$$

## Tuning of hyperparameters

Prior to obtaining the final results, a tuning process is performed to build a well-trained model using the proposed method. The optimal configuration involves utilizing a batch size of 16, and specifying the maximum number of epochs as 50. During training, smooth learning strategy is implemented, with a patience of 3 epochs and a decay factor of 0.2. This means that if there is no improvement in performance for 3 consecutive epochs, the learning rate is reduced by the decay factor. Also early stopping strategy is implemented to terminate the training early in case of overfitting. A hybrid loss function, which is a combination of dice loss and focal loss, is chosen as the loss function of the model, and the Adam optimizer is used for optimizing the network. A summary of the hyperparameters used is provided in Table 2, where $L_0$ denotes the initial learning rate.

## Results

We have experimented on the BUSI dataset on two settings. Initially, we have utilized grayscale images from the BUSI dataset as the input for the basic vanilla U-Net model, which we refer to as the GSU-Net, to generate the corresponding segmentation map. Subsequently, we have

**Table 2. Hyperparameter configurations to train the proposed model.**

| Hyperparameter | Value |
|---|---|
| Batch | 16 |
| Epoch | 50 |
| $L_0$ | 0.0001 |
| Gradient decay policy | ReduceLROnPlateau |
| Patience epoch | 3 |
| Decay factor | 0.2 |
| Optimizer | Adam |

**Table 3. Segmentation performance comparison between GSU-Net and the proposed DBU-Net on the BUSI dataset.**

| 5-Fold CV | GSU-Net | | | DBU-Net | | |
|---|---|---|---|---|---|---|
| | $Acc_p$ | IoU | DSC | $Acc_p$ | IoU | DSC |
| Fold 1 | 94.00 | 70.00 | 82.40 | 94.70 | 72.40 | 84.00 |
| Fold 2 | 94.20 | 74.90 | 85.70 | 95.60 | 76.30 | 86.50 |
| Fold 3 | 93.90 | 70.80 | 82.90 | 94.90 | 75.20 | 85.90 |
| Fold 4 | 93.90 | 71.60 | 83.50 | 94.10 | 73.00 | 84.40 |
| Fold 5 | 93.80 | 73.80 | 84.90 | 94.20 | 74.80 | 85.60 |
| Mean | 94.20 | 72.22 | 83.88 | **94.70** | **74.34** | **85.28** |
| Std. Dev. | 0.675 | 2.062 | 1.383 | **0.604** | **1.609** | **1.047** |

All values are in %, and superior results are indicated in bold style.

utilized the Roberts cross operator on the images to generate a modified image set. Finally, we have employed these two types of image as inputs for two distinct branches of our newly proposed DBU-Net model. The experiments conducted on the BUSI dataset utilize a 5-fold cross-validation technique. The detailed results of the 5 folds are presented in Table 3.

Table 3 displays the results in terms of pixel accuracy ($Acc_p$), IoU and DSC score. It shows the values obtained from each fold, as well as the mean and standard deviation values across the five folds. It can be observed from Table 3 and Fig 6 that the proposed DBU-Net achieves a higher mean IoU and DSC value compared to the GSU-Net. The overall pixel accuracy is also high in DBU-Net as compared to the GSU-Net. It clearly shows the supremacy of dual branching in segmenting the breast ultrasound images.

Fig 7 illustrates the learning curves of the proposed model, displaying the model's progress in terms of loss, accuracy, dice-coefficient, and F1-score for Fold 1 and Fold 2. We specifically showcase these two folds as they yield the minimum and maximum metric values, respectively, as observed in Table 4. The loss curve demonstrates the model's smooth learning process, while the accuracy curve showcases the training and validation accuracies as the number of epochs increases. Moreover the Dice coefficient and F1-score curve indicates the model's ability to accurately segment pixels and its overall performance throughout the training process.

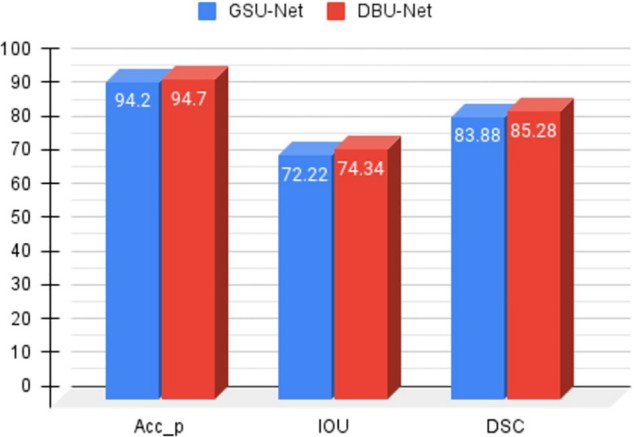

**Fig 6. Performance comparison between the GSU-Net and the DBU-Net over the mean values of 5 folds.**

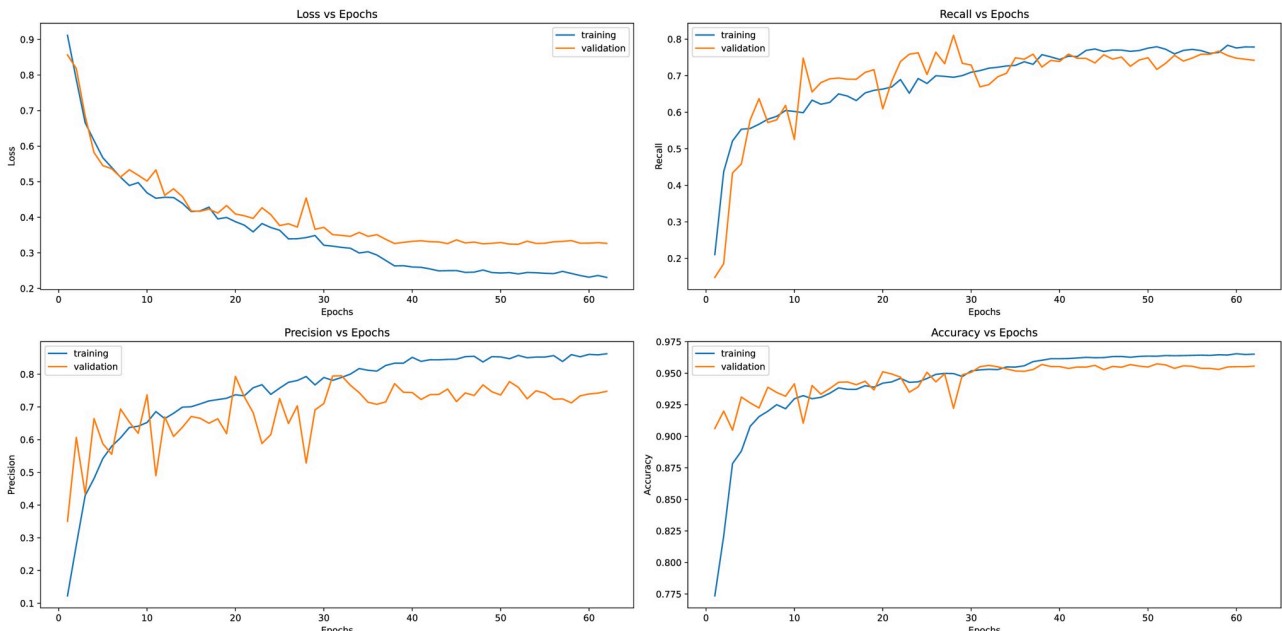

**Fig 7. Learning curves (Loss vs Epochs, Recall vs Epochs, Precision vs Epochs, and Accuracy vs Epochs) of the proposed model on Fold 1 of experiments.**

## Comparative analysis of different DBU-Nets

In this subsection, we discuss the results of a study concerning various other edge operators namely Prewitt and Sobel. The relevant results with a 5-fold cross-validation technique are tabulated under Table 4.

It can be observed from Tables 3 and 4 that the Roberts-based DBU-Net outperforms the other two DBU-Nets with a high margin. The main task of an edge operator is to capture whenever the change in pixel intensities is relatively high. In a typical edge operator, there are two components: the X operator and the Y operator. The job of these operators is to detect the presence of horizontal ($O_x$) and vertical ($O_y$) edges. The Prewitt and Sobel edge operators consider a $3 \times 3$ matrix for this task, whereas the Roberts edge operator considers a $2 \times 2$ matrix. It should also be noted that all these operators satisfy the relation $O_x^T = O_y$, which essentially means that each of these horizontal and vertical operators is rotated by 90deg relative to each

**Table 4. Performance comparison of the DBU-Net model with Prewitt and Sobel filters on the BUSI dataset.** All values are in %.

| 5-Fold CV | Prewitt based DBU-Net | | | Sobel based DBU-Net | | |
|---|---|---|---|---|---|---|
| | $Acc_p$ | IoU | DSC | $Acc_p$ | IoU | DSC |
| Fold 1 | 93.90 | 70.30 | 82.50 | 94.50 | 72.00 | 84.00 |
| Fold 2 | 95.00 | 73.60 | 84.80 | 95.50 | 75.90 | 85.90 |
| Fold 3 | 94.20 | 72.80 | 84.30 | 94.90 | 73.40 | 84.60 |
| Fold 4 | 93.60 | 69.10 | 81.70 | 93.60 | 71.70 | 83.50 |
| Fold 5 | 94.60 | 76.00 | 86.30 | 94.20 | 74.00 | 85.00 |
| **Mean** | 94.26 | 72.36 | 83.92 | 94.54 | 73.40 | 84.60 |
| **Std. Dev.** | 0.496 | 2.443 | 1.645 | 0.640 | 1.514 | 0.827 |

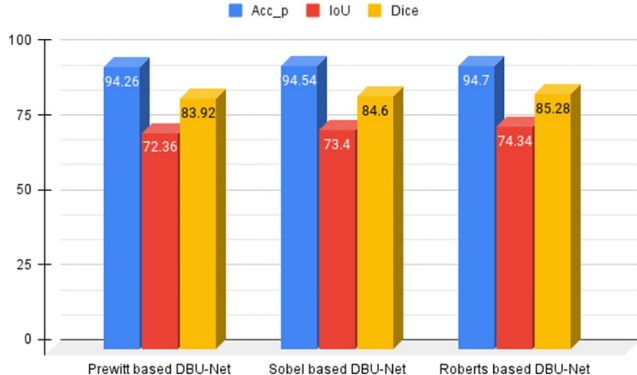

**Fig 8. Comparative study of IoU and Dice score concerning different DBU-net models based on the mean values of 5 folds.**

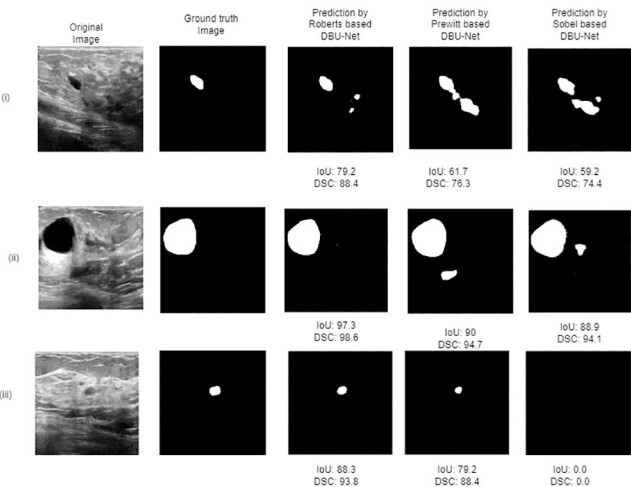

**Fig 9. Analysis of predicted segmentation masks produced by different filter based DBU-Nets.**

other. The smaller kernels in Roberts edge operator help to amplify the local edges and thus produce a better edge map. The same is reflected in the obtained results as shown in Fig 8.

We have also assessed the performance of different DBU-Nets on sample cases, as depicted in Fig 9. The results indicate that both the Prewitt-based and Sobel-based DBU-Net models generate false positive results in case (i) and case (ii), despite having high IoU and DSC values. However, the Roberts-based DBU-Net accurately predicts the segmented mask in case (i) with relatively fewer false positive regions. In case (ii), it does not produce any false positive regions. Furthermore, in case (iii), the Sobel-based DBU-Net fails to predict the segmentation mask altogether whereas Prewitt-based DBU-Net identify the actual region but with significantly lower IoU and DSC scores compared to the Roberts-based DBU-Net.

## Statistical analysis

We have performed a statistical test to assess the robustness of the proposed DBU-Net compared to the GSU-Net model. The null hypothesis states that *'there is no difference in the mean*

Table 5. Wilcoxon signed rank test results of the proposed DBU-Net model.

| Metric | p-value |
|--------|---------|
| IoU | 0.0312 |
| DSC | 0.0312 |
| $Acc_p$ | 0.0312 |

*results produced by the DBU-Net when compared to the GSU-Net'*, while the alternative hypothesis is that there is a difference in the mean results between the two models. To perform this test, we have considered the widely used non-parametric statistical test known as the Wilcoxon signed rank test [35]. We have compared the mean IoU, DSC, and $Acc_p$ values of the GSU-Net and DBU-Net models fold-wise. The obtained results are presented in Table 5. Based on the results presented in Table 5, we can clearly reject the null hypothesis for each case, as the obtained p-values are less than 0.05 (5%). Therefore, we can assert that the proposed DBU-Net produces statistically significant results.

## Analysis of the segmentation results

This subsection focuses on the results of a study on the successful and unsuccessful cases identified by the proposed method. Fig 10 illustrates some of the successful predictions made by the proposed method, along with the corresponding values of the evaluation metrics. As previously mentioned, the IoU measures the degree of overlap between two masks, and a higher IoU value indicates a more accurate prediction of the segmentation mask. The same is true for the DSC value, which also represents the accuracy of the predicted segmentation mask. It is to be noted that successful prediction is done by the proposed model for both small (first two cases) and large (last two cases) types of tumors. While the DBU-Net demonstrates impressive results in segmenting tumors of various sizes across the BUSI datasets, it encounters difficulties in detecting tumor regions properly in a few highly challenging cases, as depicted in Fig 11. These challenging cases are characterized by factors such as high speckle noise, extremely low

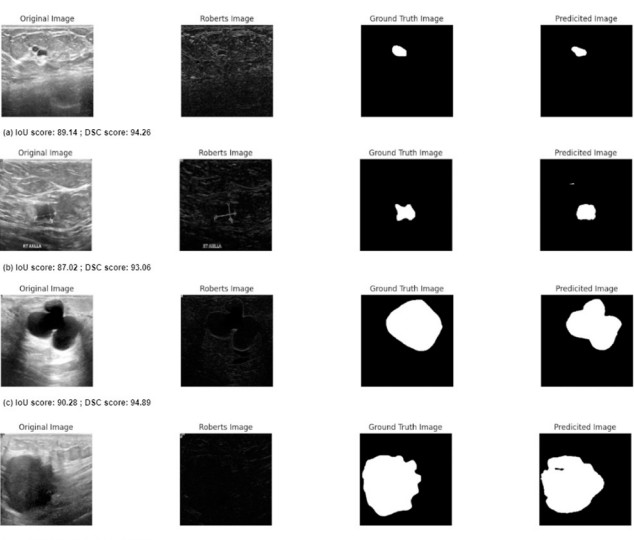

**Fig 10. Some successful predictions by the proposed model.**

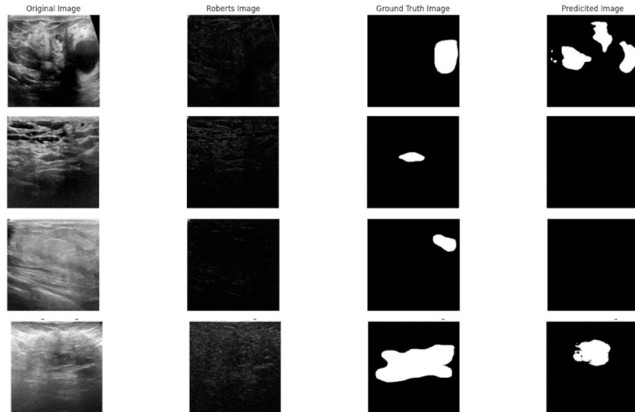

**Fig 11. Some unsuccessful predictions by the proposed model.**

contrast, and the absence of clear tumor boundaries. Despite its overall effectiveness, the DBU-Net faces limitations in handling such extreme scenarios, where the aforementioned factors posed significant challenges to accurate tumor region detection.

## Additional experiment

We have conducted an additional test to evaluate the performance of our model using only the dice loss. The results obtained from this test are presented in Table 6. From Tables 3 and 6, it can be seen that the hybrid loss, which combines both the dice loss and focal loss, has shown superior results compared to the dice loss alone. This can be attributed to the following reasons that the dice loss primarily focuses on measuring the overlap between predicted and ground truth masks. However, it may not adequately penalize false positives and false negatives. The focal loss, on the other hand, assigns higher weights to challenging samples, which helps in better localizing the object boundaries and reducing false positives/negatives. Also the dataset considered here is struggling with high class imbalance problem. The dice loss alone may not be enough to handle this issue, as it treats all classes equally. In contrast, the focal loss assigns higher weights to minority classes, effectively addressing the imbalance and improving the model's ability to learn from them.

Fig 12 demonstrates the effectiveness of utilizing a hybrid loss function for achieving superior performance over dice loss function alone.

**Table 6. Results of the proposed DBU-Net model with dice loss function using the BUSI dataset.**

| 5-Fold CV | $Acc_p$(%) | IoU(%) | DSC(%) |
|---|---|---|---|
| Fold 1 | 93.30 | 70.70 | 82.80 |
| Fold 2 | 94.80 | 73.60 | 84.80 |
| Fold 3 | 93.90 | 72.70 | 84.20 |
| Fold 4 | 93.10 | 70.50 | 82.70 |
| Fold 5 | 93.80 | 73.70 | 84.80 |
| **Mean** | 93.78 | 72.24 | 83.86 |
| **Std. Dev.** | 0.661 | 1.549 | 1.043 |

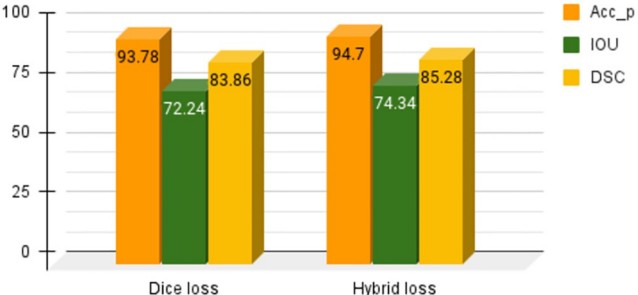

**Fig 12. Performance comparison of the proposed method using dice loss and the hybrid loss.**

## State-of-the-art comparison

Finally the performance of the proposed method is compared with other state-of-the-art (SOTA) approaches found in the literature. The results are tabulated in Table 7, which shows that our proposed method surpasses all other methods in both IoU and DSC values with a good margin. Among the methods compared in Table 7, the method used in [36] incorporates two encoders to capture and combine contextual information from images at various scales for segmentation. This method specifically demonstrates its ability to accurately identify small sized tumors. In [37, 38], attention mechanisms and transfer learning methods are employed. Whereas multi-scale guidance block and attention module are used in [39]. In [19, 40], the authors have used embedded ResNet-101 as the backbone network, whereas in [41] embedded SEResnext-50 is used as the backbone network. In [21] vanilla U-Net is used. It is to be noted that the result of [21] is obtained with the current setup. Although all the said methods, including ours, have some failure cases, the results presented in Table 7 demonstrate that our proposed method outperforms all previous methods in terms of IoU and DSC scores. This superior performance can be attributed to the dual branch method, where the use of Roberts edge operator amplifies the local edges, thereby producing a better edge map and improving the accuracy of the image segmentation of breast ultrasound images.

## Experimentation on UDIAT dataset

To assess the efficacy of our proposed DBU-Net method, we conduct an evaluation on another well-known breast ultrasound image dataset, known as the UDIAT dataset, also referred to as

**Table 7. Comparative performance analysis of DBU-Net model with SOTA approaches for BUSI dataset, highlighting superior results in bold.**

| Method | Model Name | IoU(%) | DSC(%) |
|---|---|---|---|
| Shareef et al. [36] | ESTAN | 70.00 | 78.00 |
| Xue et a. [37] | GG-Net | 73.80 | 82.10 |
| Xu et al. [38] | MSSA-Net | 71.90 | 80.65 |
| Lu et al. [39] | HAG-Net | 74.20 | 82.60 |
| Long et. al [19] | FCN | 74.00 | 82.20 |
| Chen et al. [40] | Deeplabv3+ | 73.30 | 81.90 |
| Tang et al. [41] | FPNN-TMEL | 73.60 | 81.70 |
| Ronneberger et al. [21] | U-Net | 72.22 | 83.88 |
| **Proposed Method** | **DBU-Net** | **74.34** | **85.28** |

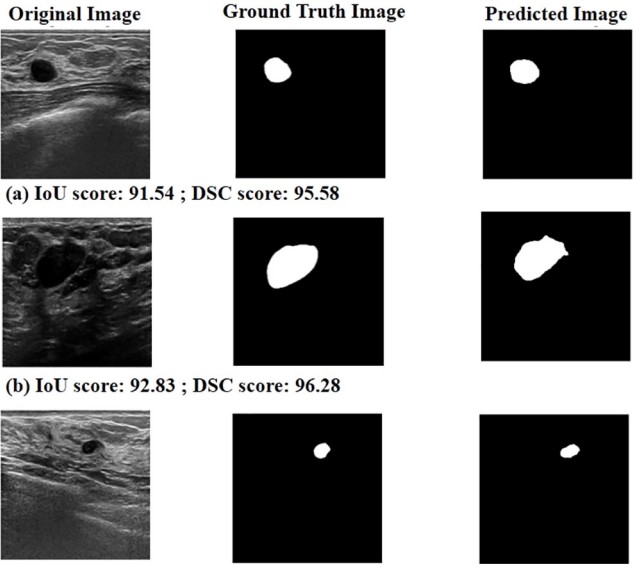

**(a) IoU score: 91.54 ; DSC score: 95.58**

**(b) IoU score: 92.83 ; DSC score: 96.28**

**(c) IoU score: 86.71 ; DSC score: 92.88**

**Fig 13. Segmentation performance of the proposed model on the UDIAT dataset.**

the Dataset B. This dataset has been generously shared by the UDIAT Diagnostic Centre in Sabadell, Spain [42] for the research purpose. It consists of a total of 109 benign and 58 malignant ultrasound images. Highly experienced radiologists meticulously labeled all regions of interest (ROIs) in these images, providing binary masks for ground truth. The average resolution of both the ultrasound images and the ground truth masks is $760 \times 570$ pixels.

For the evaluation purpose on the UDIAT dataset, we maintain the same hyperparameters that are utilized on the BUSI dataset. The segmentation results are depicted in Fig 13. Notably, Table 8 showcases the impressive quantitative performance achieved by our proposed model in comparison to prior notable research conducted on this dataset.

## Conclusion and future scope

Ultrasonography image segmentation of breast cancer is an important step toward identifying the presence of any lesion. Recently, the U-Net and its variants have proven to be very useful in segmenting various images including medical images. In the present work, we propose a new variation of the U-Net architecture, dubbed DBU-Net, for the segmentation of breast

**Table 8. Performance comparison of the proposed DBU-Net model with previous models using the UDIAT dataset.**

| Work Ref. | Model | IoU(%) | DSC(%) |
|---|---|---|---|
| Shareef et al. [36] | ESTAN | 74.00 | 82.00 |
| Xu et al. [38] | MSSA-Net | 76.05 | 83.78 |
| Shareef et al. [43] | STAN | 69.50 | 78.20 |
| Cho et al. [44] | RFS-UNet | 77.09 | 85.36 |
| **Proposed** | **DBU-Net** | **77.46** | **87.28** |

Bold values indicate superior results.

ultrasound images. Our proposed model utilizes two separate paths for the encoding process —one path takes the original image as the input, and the other path uses the Roberts edge information obtained from the original image. This dual-branch encoding scheme helps to enrich the semantic information within the latent space, and also helps to cross-learning. To allow the cross-learning, we have applied a weighted addition mechanism, while the weight is decided based on the loss gradient of the training of the model. The performance of the DBU-Net model is evaluated using the BUSI and UDIAT datasets, yielding IoU scores of 74.34 and 77.46, and DSC scores of 85.28 and 87.28, respectively. Though the obtained results are encouraging, however, to make it applicable for the practical cases, it should be completely error-free. False positives or false negatives can lead to wrong diagnosis. Therefore, in the future, we plan to take care of these issues. We may apply some image pre-processing techniques like denoising to remove unwanted noise from image to analyze it in better form and to feed better information to the deep model. To improve the performance of the model, attention mechanisms like Convolutional Block Attention Module (CBAM) and Shifting Window Attention (SWA) can be applied. We can think of applying a different operation to aid the cross-learning strategy. Also, to ensure the robustness of the model, we will evaluate our segmentation model on other modality of medical image applications.

## Acknowledgments

We are thankful to the Center for Microprocessor Applications for Training Education and Research (CMATER) research laboratory of the Computer Science and Engineering Department, Jadavpur University, Kolkata, India, for providing infrastructural support to this research project.

## Author Contributions

**Conceptualization:** Payel Pramanik, Rishav Pramanik.

**Data curation:** Payel Pramanik.

**Formal analysis:** Friedhelm Schwenker.

**Funding acquisition:** Ram Sarkar.

**Methodology:** Rishav Pramanik, Friedhelm Schwenker.

**Project administration:** Ram Sarkar.

**Software:** Payel Pramanik.

**Supervision:** Friedhelm Schwenker, Ram Sarkar.

**Validation:** Friedhelm Schwenker, Ram Sarkar.

**Writing – original draft:** Payel Pramanik, Rishav Pramanik, Ram Sarkar.

**Writing – review & editing:** Payel Pramanik, Rishav Pramanik, Friedhelm Schwenker, Ram Sarkar.

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
