## [Decision Letter · Decision Letter 0]

4 Aug 2023

PONE-D-23-21799DBU-Net: Dual Branch U-Net for Breast Ultrasound Image SegmentationPLOS ONE

Dear Dr. Schwenker,

Thank you for submitting your manuscript to PLOS ONE. After careful consideration, we feel that it has merit but does not fully meet PLOS ONE’s publication criteria as it currently stands. Therefore, we invite you to submit a revised version of the manuscript that addresses the points raised during the review process.

We look forward to receiving your revised manuscript.

Kind regards,

Saddam Hussain Khan

Academic Editor

PLOS ONE

Journal Requirements:

Additional Editor Comments:

Please implement the proposed technique on UDIAT and other breast cancer ultrasound images to enhance the dataset's robustness and make it available to researchers, thereby accelerating research endeavors.

Reviewers' comments:

Reviewer's Responses to Questions

**Comments to the Author**

1. Is the manuscript technically sound, and do the data support the conclusions?

Reviewer #1: Partly

Reviewer #2: Partly

2. Has the statistical analysis been performed appropriately and rigorously? 

Reviewer #1: No

Reviewer #2: Yes

3. Have the authors made all data underlying the findings in their manuscript fully available?

Reviewer #1: Yes

Reviewer #2: Yes

4. Is the manuscript presented in an intelligible fashion and written in standard English?

Reviewer #1: No

Reviewer #2: Yes

5. Review Comments to the Author

Reviewer #1: 1. The abstract lacks clarity regarding the novelty of the proposed approach, omits details on validation techniques and empirical results, and requires improvement in grammatical and technical English to enhance readability.

2. The quality of figures needs to be enhanced for better readability and comprehension.

3. The sentence structure needs improvement to avoid repetition of similar words, particularly when referring to table names.

4. The novelty of the proposed technique based on existing deep UNet CNN should be elaborated in detail, providing a rationale for the approach and discussing the potential impact of the techniques.

5. To compute the results, it is suggested to use UDIAT in conjunction with the BUS dataset. Additionally, making the datasets public would facilitate further research activities.

6. A detailed ablation study, including results and computational complexity, should be provided to better understand the effectiveness and efficiency of the proposed method.

7. Enhancing the novelty of the paper can be achieved by considering recent medical challenges and incorporating technical innovations and concepts in deep segmentation CNN may be considering the below manuscripts:

a. Zafar, Muhammad Mohsin, Zunaira Rauf, Anabia Sohail, Abdul Rehman Khan, Muhammad Obaidullah, Saddam Hussain Khan, Yeon Soo Lee, and Asifullah Khan. "Detection of tumour infiltrating lymphocytes in CD3 and CD8 stained histopathological images using a two-phase deep CNN." Photodiagnosis and Photodynamic Therapy 37 (2022): 102676..

b. Khan, Saddam Hussain, Asifullah Khan, Yeon Soo Lee, Mehdi Hassan, and Woong Kyo Jeong. "Segmentation of shoulder muscle MRI using a new region and edge based deep auto-encoder." Multimedia Tools and Applications 82, no. 10 (2023): 14963-14984.

c. Zahoor, Mirza Mumtaz, and Saddam Hussain Khan. "Brain tumor MRI Classification using a Novel Deep Residual and Regional CNN." arXiv preprint arXiv:2211.16571 (2022).

d. Khan, Saddam Hussain. "COVID-19 Detection and Analysis From Lung CT Images using Novel Channel Boosted CNNs." arXiv preprint arXiv:2209.10963 (2022).

e. Khan, Saddam Hussain, Najmus Saher Shah, Rabia Nuzhat, Abdul Majid, Hani Alquhayz, and Asifullah Khan. "Malaria parasite classification framework using a novel channel squeezed and boosted CNN." Microscopy 71, no. 5 (2022): 271-282.

Reviewer #2: The authors propose a paper on “DBU-Net: Dual Branch U-Net for Breast Ultrasound Image Segmentation". I believe that the manuscript is well organized and explained. Following comments can improve its quality.

1. The authors should follow this order for your revised abstract: (1) Novelty (2) Methods (3) Results (the most noticeable numbers) (4) conclusion. Results are of high significance, and they need to provide the most important outcomes here. Please revise the abstract accordingly.

2. Results ratio miss and also cross validation

3. Introduction section is weak. I would recommend to split it in problem, objectives, literature, and contribution.

4. Is this model applicable for color model if yes then why not use it

5. Please, report the recent medical image analysis challenges. Moreover, the paper can be strengthened with technical and terminology innovations by may be considering the below recent manuscripts:

1. Rauf, Zunaira, Anabia Sohail, Saddam Hussain Khan, Asifullah Khan, Jeonghwan Gwak, and Muhammad Maqbool. "Attention-guided multi-scale deep object detection framework for lymphocyte analysis in IHC histological images." Microscopy (2022).

2. Zahoor, M.M.; Qureshi, S.A.; Bibi, S.; Khan, S.H.; Khan, A.; Ghafoor, U.; Bhutta, M.R. A New Deep Hybrid Boosted and Ensemble Learning-Based Brain Tumor Analysis Using MRI. Sensors 2022, 22, 2726. https://doi.org/10.3390/s22072726

3. Khan, Saddam Hussain, Javed Iqbal, Syed Agha Hassnain, Muhammad Owais, Samih M. Mostafa, Myriam Hadjouni, and Amena Mahmoud. "Covid-19 detection and analysis from lung ct images using novel channel boosted cnns." Expert Systems with Applications 229 (2023): 120477.

4. Khan, Saddam Hussain, Najmus Saher Shah, Rabia Nuzhat, Abdul Majid, Hani Alquhayz, and Asifullah Khan. "Malaria parasite classification framework using a novel channel squeezed and boosted CNN." Microscopy 71, no. 5 (2022): 271-282.

6. Are there any methodology limitations and/or challenges required to be addressed? All these should be mentioned at the end of the Conclusions section.

7. Why you select relu function as activation function why not Sigmoid, Tanh, ReLU, Leaky ReLU, PReLU, ELU, and SELU. Do compression in form of table with other activation function.

8. In data set you mentioned three classes how you make classes normal, benign, and malignant.

9. Uneven dataset it will give False positive

10. Fig 5 and 7 are not visible please update them with readable legends.

11. What is the author’s contribution? Please include paragraph with comparison with recent literature.

12. ROC and confusion matrix

13. Contribution section must be cited with recent paper what novel aspect authors achieved.

14. Table 7 must be elaborated how these different model lacks what’s author’s criteria?

15. All equations must be cited if are not authors work.

16. Future direction are too generic.

6. PLOS authors have the option to publish the peer review history of their article (what does this mean?). If published, this will include your full peer review and any attached files.

Reviewer #1: No

Reviewer #2: No

---

## [Author Response · Author response to Decision Letter 0]

15 Sep 2023

We have no special information to the revision besides the response letter with detailed comments to the reviewers' comments.

---

## [Decision Letter · Decision Letter 1]

17 Oct 2023

DBU-Net: Dual Branch U-Net for Breast Ultrasound Image Segmentation

PONE-D-23-21799R1

Dear Dr. Schwenker,

We’re pleased to inform you that your manuscript has been judged scientifically suitable for publication and will be formally accepted for publication once it meets all outstanding technical requirements.

Kind regards,

Saddam Hussain Khan

Academic Editor

PLOS ONE

Additional Editor Comments (optional):

Reviewers' comments:

Reviewer's Responses to Questions

**Comments to the Author**

1. If the authors have adequately addressed your comments raised in a previous round of review and you feel that this manuscript is now acceptable for publication, you may indicate that here to bypass the “Comments to the Author” section, enter your conflict of interest statement in the “Confidential to Editor” section, and submit your "Accept" recommendation.

Reviewer #2: All comments have been addressed

Reviewer #3: All comments have been addressed

2. Is the manuscript technically sound, and do the data support the conclusions?

Reviewer #2: Yes

Reviewer #3: Yes

3. Has the statistical analysis been performed appropriately and rigorously? 

Reviewer #2: Yes

Reviewer #3: Yes

4. Have the authors made all data underlying the findings in their manuscript fully available?

Reviewer #2: Yes

Reviewer #3: Yes

5. Is the manuscript presented in an intelligible fashion and written in standard English?

Reviewer #2: Yes

Reviewer #3: Yes

6. Review Comments to the Author

Reviewer #2: The challenges within the Breast Ultrasound dataset are of great interest. I believe that discussing these challenges in more detail within your paper would be highly beneficial. It would provide context for researchers using the dataset and help them understand potential limitations. Additionally, if you have insights or recommendations on how to address these challenges, sharing them would be valuable.

Reviewer #3: Firstly, I wanted to commend you on providing access to the Breast Ultrasound dataset. However, it appears that the link provided in your publication is not working. It would be immensely helpful if you could verify the link or provide an alternative source for accessing the dataset. Accessible data is crucial for the scientific community to replicate and build upon your research.

7. PLOS authors have the option to publish the peer review history of their article (what does this mean?). If published, this will include your full peer review and any attached files.

Reviewer #2: No

Reviewer #3: No

---

## [Editor Report · Acceptance letter]

26 Oct 2023

PONE-D-23-21799R1 

DBU-Net: Dual Branch U-Net for Tumor Segmentation in Breast Ultrasound Images 

Dear Dr. Schwenker:

I'm pleased to inform you that your manuscript has been deemed suitable for publication in PLOS ONE. Congratulations! Your manuscript is now with our production department. 

Kind regards, 

on behalf of

Dr. Saddam Hussain Khan 

Academic Editor

PLOS ONE